EMBO
Molecular Medicine

# A clinical and mechanistic study of topical borneol-induced analgesia

Shu Wang[1,2,*,†] iD, Dan Zhang[3,†], Jinsheng Hu[1,2,†], Qi Jia[3,†], Wei Xu[3], Deyuan Su[1,2], Hualing Song[4], Zhichun Xu[1,2], Jianmin Cui[1,5], Ming Zhou[1,6], Jian Yang[1,7,**] iD & Jianru Xiao[3,***] iD

## Abstract

Bingpian is a time-honored herb in traditional Chinese medicine (TCM). It is an almost pure chemical with a chemical composition of (+)-borneol and has been historically used as a topical analgesic for millennia. However, the clinical efficacy of topical borneol lacks stringent evidence-based clinical studies and verifiable scientific mechanism. We examined the analgesic efficacy of topical borneol in a randomized, double-blind, placebo-controlled clinical study involving 122 patients with postoperative pain. Topical application of borneol led to significantly greater pain relief than placebo did. Using mouse models of pain, we identified the TRPM8 channel as a molecular target of borneol and showed that topical borneol-induced analgesia was almost exclusively mediated by TRPM8, and involved a downstream glutamatergic mechanism in the spinal cord. Investigation of the actions of topical borneol and menthol revealed mechanistic differences between borneol- and menthol-induced analgesia and indicated that borneol exhibits advantages over menthol as a topical analgesic. Our work demonstrates that borneol, which is currently approved by the US FDA to be used only as a flavoring substance or adjuvant in food, is an effective topical pain reliever in humans and reveals a key part of the molecular mechanism underlying its analgesic effect.

**Keywords** borneol; pain; topical analgesic; traditional Chinese medicine; TRPM8
**Subject Categories** Neuroscience; Pharmacology & Drug Discovery

## Introduction

For treatment of acute and chronic pain, analgesics, such as non-steroidal anti-inflammatory drugs (NSAIDs) and opioids, are typically administered systemically (Sawynok, 2003; Argoff, 2013). However, systemic analgesics frequently cause adverse reactions, which limits their usefulness (Sawynok, 2003; Argoff, 2013). For example, systemic NSAIDs carry common risks for the development of significant adverse effects on the gastrointestinal tract and hepatic, renal, and central nervous systems (Clive & Stoff, 1984; Roth, 1988; Hoppmann et al, 1991; Murray & Brater, 1993; Garcia Rodriguez & Jick, 1994; Boelsterli, 2002; Laporte et al, 2004; Rostom et al, 2005), and systemic opioid analgesics can result in potentially fatal respiratory depression, addiction, nausea, pruritus, and constipation (Friedman & Dello Buono, 2001; Compton & Volkow, 2006; Paulozzi et al, 2006; Porreca & Ossipov, 2009; Dahan et al, 2010). These adverse effects have stimulated interests in localized, non-systemic approaches to manage pain. Topical analgesics are pain relievers in externally applied formulations, such as sprays, creams, gels, plasters, or patches, and are applied locally to the skin of the pain area. Topical treatments offer the advantage of local, enhanced analgesic delivery to affected tissues with lower incidences of systemic adverse effects due to reduced plasma concentrations (Sawynok, 2003; Argoff, 2004, 2013; de Leon-Casasola, 2007; McCleane, 2007). In addition, the use of topical analgesics also has a low risk for drug–drug interactions, which is especially important in the elderly and in those who are on multiple medications (Cavalieri, 2002; Sawynok, 2003; Argoff, 2004; de Leon-Casasola, 2007). Other advantages of topical analgesics include the lack of need to titrate doses based on tolerability and, importantly, their ease of use (Sawynok, 2003; Argoff, 2004). Currently, numerous clinical trials of topical analgesics are ongoing. However, only a limited number of active agents are available as topical analgesics on the market,

1   Key Laboratory of Animal Models and Human Disease Mechanisms of Chinese Academy of Sciences/Key Laboratory of Bioactive Peptides of Yunnan Province, Ion Channel Research and Drug Development Center, Kunming Institute of Zoology, Chinese Academy of Sciences, Kunming, China
2   Kunming College of Life Science, University of Chinese Academy of Sciences, Kunming, China
3   Department of Orthopedic Oncology, Shanghai Changzheng Hospital, The Second Military Medical University, Shanghai, China
4   Department of Preventive Medicine, Shanghai University of Traditional Chinese Medicine, Shanghai, China
5   Department of Biomedical Engineering, Center for the Investigation of Membrane Excitability Disorders, Cardiac Bioelectricity and Arrhythmia Center, Washington University, St. Louis, MO, USA
6   Verna and Marrs McLean Department of Biochemistry and Molecular Biology, Baylor College of Medicine, Houston, TX, USA
7   Department of Biological Sciences, Columbia University, New York, NY, USA
    *Corresponding author. Tel: +86 0871 65189257; E-mail: wangshu@mail.kiz.ac.cn or shu_wang@outlook.com
    **Corresponding author. Tel: +1 212 8546161; E-mail: jy160@columbia.edu or jianyang@mail.kiz.ac.cn
    ***Corresponding author. Tel: +86 021 81886843; E-mail: jianruxiao83@163.com
    †These authors contributed equally to this work

mainly including NSAIDs, capsaicin, local anesthetics, and menthol (Sawynok, 2003; de Leon-Casasola, 2007; McCleane, 2007; Argoff, 2013). Due to the diversity of the mechanisms involved in pain signaling, individual topical analgesic agents that target one particular mediator may only work for certain pain conditions or relieve pain partially (Sawynok, 2003). Thus, there is great interest in identifying new topical analgesics targeting new pathways that may serve as an alternative or supplement to systemic analgesic therapies.

Traditional Chinese medicine has been widely used in East Asia for thousands of years to provide treatments and cures for diseases, including pathological pain. However, traditional herbal remedies have encountered substantial skepticism from Western medical professionals and scientists because "the claims made for them rest largely on anecdotes and clinical observations instead of randomized, double-blind, placebo-controlled trials" and the lack of verifiable scientific basis (Yuan & Lin, 2000; Normile, 2003; Chan, 2014; Leshner, 2014). This is also the case with bingpian. Bingpian, also known as longnao in Chinese, is a time-honored herb in TCM and has been used for more than 2,000 years in clinical applications. Natural bingpian is the resin obtained from *Cinnamomum* tree and is almost a pure chemical (purity > 96%; Chinese Pharmacopoeia Commission, 2015). Its chemical composition has been identified as (+)-borneol, which is a bicyclic monoterpene (Chinese Pharmacopoeia Commission, 2015). Because borneol can be chemically synthesized from turpentine oil or camphor, synthetic bingpian is also commonly used in China (Chinese Pharmacopoeia Commission, 2015). Thus, "borneol" is now synonymous with "bingpian". According to ancient medical literature and the Chinese Pharmacopoeia, borneol can reduce pain and swelling (Li, 1578; Wang, 1758; Luo, 1789; Chinese Pharmacopoeia Commission, 2015),and it is used in greater frequency for topical application in the treatment of injuries, burns, sprains, muscle pain, hemorrhoids, carbuncles, joint pain, sore gums, and ulcerations. In addition, borneol may facilitate the delivery of other effective components and enhance their effects in combinatorial herbal formulas (Kou, 1116; Li, 1578). Thus, borneol is also orally administered as an adjuvant component for the treatment of a variety of diseases, and the recommended oral dosing of natural borneol is 0.3–0.9 g/day for adults (Chinese Pharmacopoeia Commission, 2015). Although borneol has been used successfully for millennia, few clinical studies that meet international quality standards (randomized double-blind and placebo-controlled) have been performed to demonstrate its clinical efficacy. To our knowledge, no completed or ongoing high-quality clinical studies were designed to examine the analgesic effect of topical borneol. Borneol is currently approved by the US FDA to be used only as a flavoring substance or adjuvant in food (21 CFR 172.515). Thus, the analgesic effect of topical borneol remains to be established through rigorous clinical studies.

Another difficult issue in TCM-based drug development involves deciphering the scientific mechanisms responsible for the clinical effects. At the onset of our study, three molecular targets of borneol had been identified in mammals, the $GABA_A$ receptor and TRPA1 and TRPV3 channels. Borneol can potentiate ($EC_{50} = 248$ μM) or directly activate (> 1.5 mM) the $GABA_A$ receptor (Granger *et al*, 2005), activate the TRPV3 channel ($EC_{50} = 3.45 \pm 0.13$ mM; Vogt-Eisele *et al*, 2007), and inhibit the TRPA1 channel ($IC_{50} = 0.5 \pm 0.3$ or $0.20 \pm 0.06$ mM; Takaishi *et al*, 2014;

Sherkheli *et al*, 2015). Whether these targets contribute to the analgesic effect of topical borneol has not been determined. In the past decades, the analgesic effects of borneol have been experimentally verified in various animal models (Hou *et al*, 1995; Sun *et al*, 2007; Zhao *et al*, 2010; Almeida *et al*, 2013; Jiang *et al*, 2015; Zhou *et al*, 2016), but the mechanism of action has rarely been studied and remains controversial. Moreover, in some animal studies, borneol was administered systemically or intrathecally (Jiang *et al*, 2015; Zhou *et al*, 2016); thus, these studies do not provide pertinent information to the analgesic effect of topical borneol.

In this study, we first demonstrated the clinical analgesic efficacy of topical borneol in a randomized, double-blind, placebo-controlled study involving 122 patients. We then clarified the underlying mechanism in mouse models. We found that the TRPM8 channel was activated by borneol ($EC_{50} = 65$ μM) and this activation was responsible for most of the topical borneol-induced analgesia in mice. TRPM8 is a non-selective, $Ca^{2+}$-permeable cation channel belonging to the transient receptor potential (TRP) ion channel superfamily and is mainly expressed in peripheral sensory neurons. TRPM8 is activated by cold temperature, menthol, and icillin (Julius, 2013), and its activation can elicit analgesia (Proudfoot *et al*, 2006; Dhaka *et al*, 2007; Liu *et al*, 2013). By comparing the actions of topical borneol and menthol, we show the mechanistic differences between borneol- and menthol-induced analgesia, and demonstrate that borneol exhibits advantages over menthol as a topical analgesic.

## Results

### A randomized, double-blind, placebo-controlled clinical study of topical borneol for the treatment of postoperative pain

To verify the analgesic effect of topical borneol in humans, we conducted a randomized, double-blind, placebo-controlled clinical study. Figure 1 outlines the flow of patients through this study. In total, 126 patients with postoperative pain were enrolled and randomized into the placebo control group and the borneol group. Two patients were excluded due to wound infection and a lack of clear consciousness, respectively. Another two patients quit the study without providing a reason. The remaining 60 patients in the

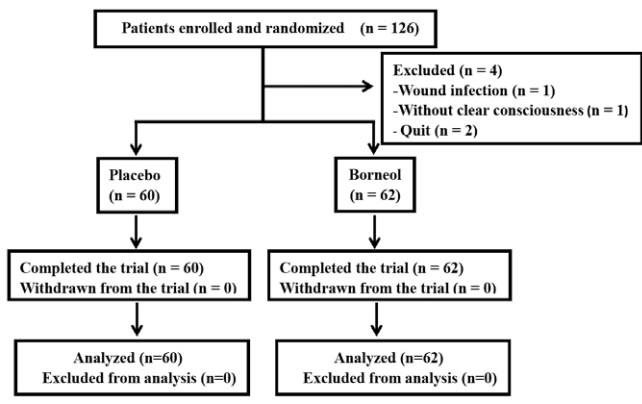

**Figure 1.  Patient flow through the clinical study**.

placebo group and 62 patients in the borneol group completed the assessment and were included in the analysis. The demographic information of the participants is presented in Table 1. Predictably, the pain intensity in the borneol group was similar to that in the placebo group before treatment. The averaged pain score in the visual analog scale (VAS) was 62.8 mm for the borneol group and 62.3 mm for the placebo group ($P = 0.8167$; Table 1). However, after a single treatment for 30–60 min, there was a significant difference in the pain score between the borneol and placebo groups ($P = 0.0004$; Table 1), and the patients in the borneol group showed significantly greater pain relief compared to the placebo group. The mean reduction in the VAS score was 32.0 in the borneol group and 19.6 in the placebo group ($P = 0.0001$; Table 1). Furthermore, 66% of patients in the borneol group experienced a ≥ 50% reduction in pain intensity compared with 35% of patients in the placebo group ($P = 0.0006$; Table 1). The relative benefit value of topical borneol was calculated to be 1.89 (95% CI 1.28–2.79) compared with placebo for the ≥ 50% reduction in pain. No adverse effects were reported by patients or observed by medical staff in this study. This clinical study indicates that topical borneol provides a significant analgesic effect in patients with postoperative pain, confirming its action as a topical analgesic in TCM.

**Topical borneol causes analgesia in mice**

To elucidate the mechanisms underlying topical borneol-induced analgesia, our study moved from bedside to bench. We first examined whether topical borneol could cause analgesia in mice as it does in humans. Intraplantar injection of capsaicin in mice produced substantial nociceptive behavior within 5 min after the injection, including lifting and licking of the injected hindpaw. Twenty-five percent medical borneol topically applied to the hindpaw prior to capsaicin injection substantially suppressed capsaicin-induced pain (Fig 2A). Topical application of 25% chemical borneol, which was obtained from Sigma-Aldrich Co., showed similar analgesic effects in the capsaicin model (Fig 2B). We also tested the effect of topical borneol on the formalin model, which is another popular animal model of acute pain. Intraplantar injection of 1.5% formalin into a mouse hindpaw produced a biphasic pain response over a test period of 60 min characterized by lifting and licking of the injected paw (Fig 2C). The pain response in the first phase occurred typically within 5 min following formalin injection, and after a quiescent period, the second phase of the pain response started and lasted for approximately 20–40 min. Topical application of borneol to the hindpaw prior to formalin injection caused

dose-dependent analgesia in both phases of the formalin-induced pain responses (Fig 2C). Furthermore, we examined the effect of topical borneol on complete Freund's adjuvant (CFA)-induced hyperalgesia. Intraplantar injection of CFA emulsion into mouse hindpaw produced inflammation, which lasted for approximately 1–2 weeks and was associated with mechanical and thermal hyperalgesia, mimicking more persistent pain and hyperalgesia in patients. Topical application of 15% borneol onto the CFA-injected hindpaw prior to pain measurement substantially attenuated mechanical and thermal hyperalgesia in mice (Fig 2D and E). Thus, topical borneol provides analgesic effects in three different mouse models of pain.

**Topical borneol-induced analgesia is largely independent of TRPA1 and GABA$_A$**

We next investigated the molecular mechanisms that underlie the analgesic effect of topical borneol in mice. We first revisited the most likely targets, TRPA1 and GABA$_A$. Inhibition of peripheral TRPA1 channel or potentiation of the spinal GABA$_A$ receptor can cause analgesia (Munro et al, 2009; Julius, 2013), and both TRPA1 and GABA$_A$ have been reported to be the molecular targets of borneol (Granger et al, 2005; Takaishi et al, 2014; Sherkheli et al, 2015) and are thought to mediate the analgesic effect of systemically or intrathecally administered borneol (Jiang et al, 2015; Zhou et al, 2016). In the capsaicin model, topical application of 15% borneol decreased the pain responses by 62% in TRPA1$^{-/-}$ mice (Fig 3A) and 85% in wild-type (WT) mice (Fig 3B). In the CFA model, topical borneol caused similar analgesia in TRPA1$^{-/-}$ mice compared with WT mice in the hot plate test (compare Figs 3C and 2E). These results suggest that TRPA1 only minimally contributes to the analgesic effect of topical borneol in mice. Intrathecal injection of the GABA$_A$ antagonist bicuculline did not show any effects on topical borneol-induced analgesia in the capsaicin model (Fig 3D and E). However, in a parallel positive control experiment, bicuculline effectively attenuated GABA$_A$ agonist muscimol-induced analgesia (Fig 3F). These results indicate that a major portion of topical borneol-induced analgesia does not function through TRPA1- or GABA$_A$-mediated mechanisms.

**TRPM8 is a molecular target of borneol**

We further examined whether borneol affects other known peripheral molecular targets in pain signaling pathways. We tested the effects of borneol on whole-cell currents of recombinant TRPV1,

**Table 1. Patient demographic and clinical characteristics.**

|  | Placebo ($n = 60$) | Borneol ($n = 62$) | P |
|---|---|---|---|
| Age | 54.73 (16.00, 50.60–58.87) | 53.34 (14.05, 49.77–56.91) | 0.4138 |
| Gender (Female/Male) | 27/33 | 25/37 | 0.6015 |
| VAS score (Before treatment) | 62.33 (15.66, 58.29–66.38) | 62.82 (15.11, 58.98–66.66) | 0.8167 |
| VAS score (After treatment) | 42.75 (19.21, 37.79–47.71) | 30.81 (13.46, 27.39–34.23) | 0.0004 |
| Absolute change in VAS score | 19.58 (17.33, 15.11–24.06) | 32.02 (14.97, 28.21–35.82) | 0.0001 |
| % change in VAS score | 30.81 (29.07, 23.30–38.32) | 50.05 (21.02, 44.71–55.38) | 0.0001 |
| % patients with a ≥ 50% pain relief | 35.00 | 66.13 | 0.0006 |

Data are expressed as mean (SD, 95% CI) unless specified.

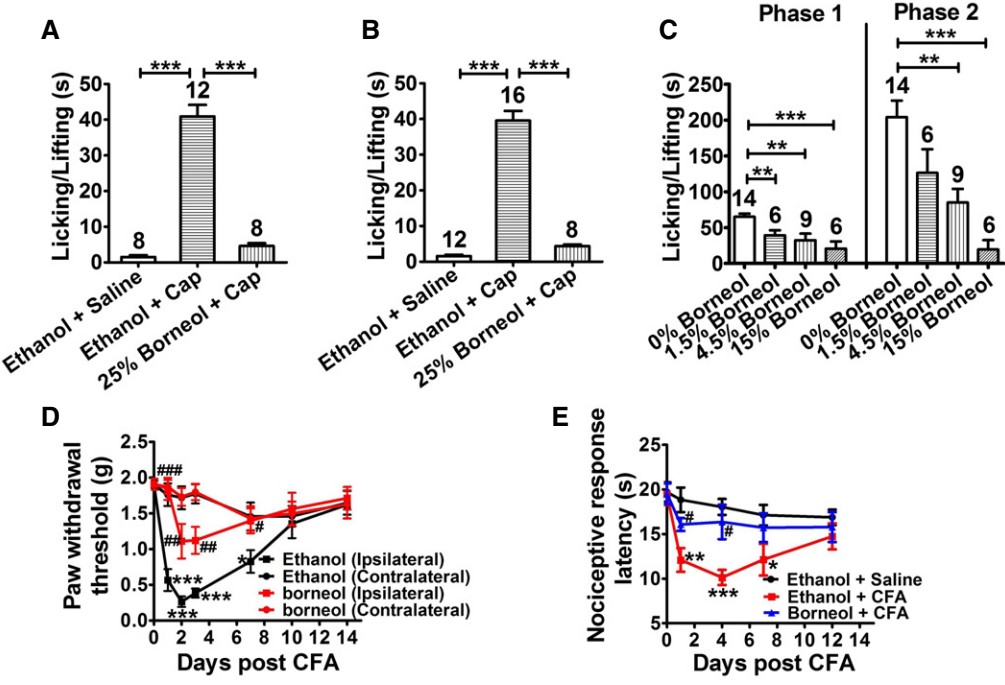

**Figure 2.  The effect of topical borneol on mouse models of pain.**

A, B    Quantification of the borneol effect on capsaicin (Cap)-induced nociceptive responses in mice. 25% of either medical borneol (A) or chemical borneol (B) was applied to a hindpaw for a total of three times at 10-min intervals, and 100 μM Cap was injected into the paw after 10 min. Paw licking and lifting time was measured within 5 min after Cap injection. Saline was used as a control to Cap, and ethanol was used as a control to borneol.

C    Quantification of the borneol effect on formalin-induced biphasic pain responses in mice. 0–15% borneol was applied to a hindpaw for a total of three times at 10-min intervals, and 1.5% formalin was injected into the paw after 10 min. Paw licking and lifting time was measured within the first 5 min (Phase 1) and within 15–60 min (Phase 2).

D    Quantification of the borneol effect on CFA-induced mechanical hyperalgesia in mice. CFA was injected into a hindpaw, and saline was injected into another hindpaw at day 0. At each experimental day, ethanol (black) or 15% borneol (red) was topically applied to both hindpaws for a total of three times at 10-min intervals. After 10 min, 50% paw withdrawal thresholds were measured with von Frey filaments in both the ipsilateral (CFA-injected) and contralateral hindpaws. Ethanol was used as a control to borneol. The statistical analysis was performed between the groups of Ethanol (Ipsilateral) and Ethanol (Contralateral) with the statistical significance indicated by asterisks, or of Borneol (Ipsilateral) and Ethanol (Ipsilateral) with the statistical significance indicated by pound signs. $n = 8$ mice for each group.

E    Quantification of the borneol effect on CFA-induced thermal hyperalgesia in mice. Either saline or CFA was injected into both hindpaws at day 0. At each experimental day, ethanol or 15% borneol was applied to both hindpaws for a total of three times at 10-min intervals. After 10 min, the nociceptive response latencies of mice were measured in a hot plate test (52°C). Saline was used as a control to CFA, and ethanol was used as a control to borneol. The statistical analysis was performed between the groups of Ethanol + CFA ($n = 9$) and Ethanol + Saline ($n = 10$) with the statistical significance indicated by asterisks, or of Borneol + CFA ($n = 10$) and Ethanol + CFA ($n = 9$) with the statistical significance indicated by pound signs.

Data information: In (A–C), the number of mice is indicated on top of each bar. Statistical significance was evaluated using two-tailed *t*-test (for two-group comparisons) or one-way analysis of variance (ANOVA) followed by Tukey's test (for multi-group comparisons). *$P < 0.05$; **$P < 0.01$; ***$P < 0.001$; #$P < 0.05$; ##$P < 0.01$; ###$P < 0.001$; the exact *P*-values are indicated in Appendix Table S1. All the data are presented as the mean ± standard error of the mean (SEM).

ASIC3, P2X2, and P2X4 expressed in HEK 293 cells (Fig 4A–D) and the enzymatic activity of cyclooxygenase-2 (COX-2; Fig 4E; Seibert *et al*, 1994; Deval *et al*, 2008; Gum *et al*, 2012; Julius, 2013). Borneol showed no significant effect on these targets. We also tested borneol effects on peripheral sensory neurons and examined whether borneol could affect high-potassium-induced excitation of dorsal root ganglion (DRG) neurons. Surprisingly, 200 μM (~0.003%) borneol did not reduce high-potassium-induced intracellular $Ca^{2+}$ increase; instead, it actually increased the $Ca^{2+}$ signal in a subset of DRG neurons, most of which (71%) also responded to menthol (Fig 4F). Menthol is well known for its ability to activate TRPM8 receptor in sensory neurons, which is a non-selective, $Ca^{2+}$-permeable cation channel (McKemy *et al*, 2002; Peier *et al*, 2002). This result suggests that TRPM8 mediates the borneol-induced

intracellular $Ca^{2+}$ increase in DRG neurons. To test this hypothesis, we examined the effect of borneol on DRG neurons from TRPM8$^{-/-}$ mice. Indeed, none of the neurons responded to borneol (Fig EV1), confirming that TRPM8 was the mediator. However, it was reported that borneol had no significant effects on TRPM8 (Vogt-Eisele *et al*, 2007). To clarify these contradicting findings, we directly tested the effect of borneol on human TRPM8 (hTRPM8) expressed in HEK 293 cells. Borneol induced an intracellular $Ca^{2+}$ increase in hTRPM8-expressing cells in a concentration-dependent manner (Fig 4G) but not in mock-transfected cells (Fig EV2A). Compared to menthol ($EC_{50} = 13$ μM) (Figs EV2B and 4H), borneol exhibited a reduced potency ($EC_{50} = 65$ μM) but similar efficacy in activating TRPM8 (Fig 4H). In addition, both menthol and borneol elicited whole-cell currents in HEK 293 cells expressing hTRPM8 (Figs 4I

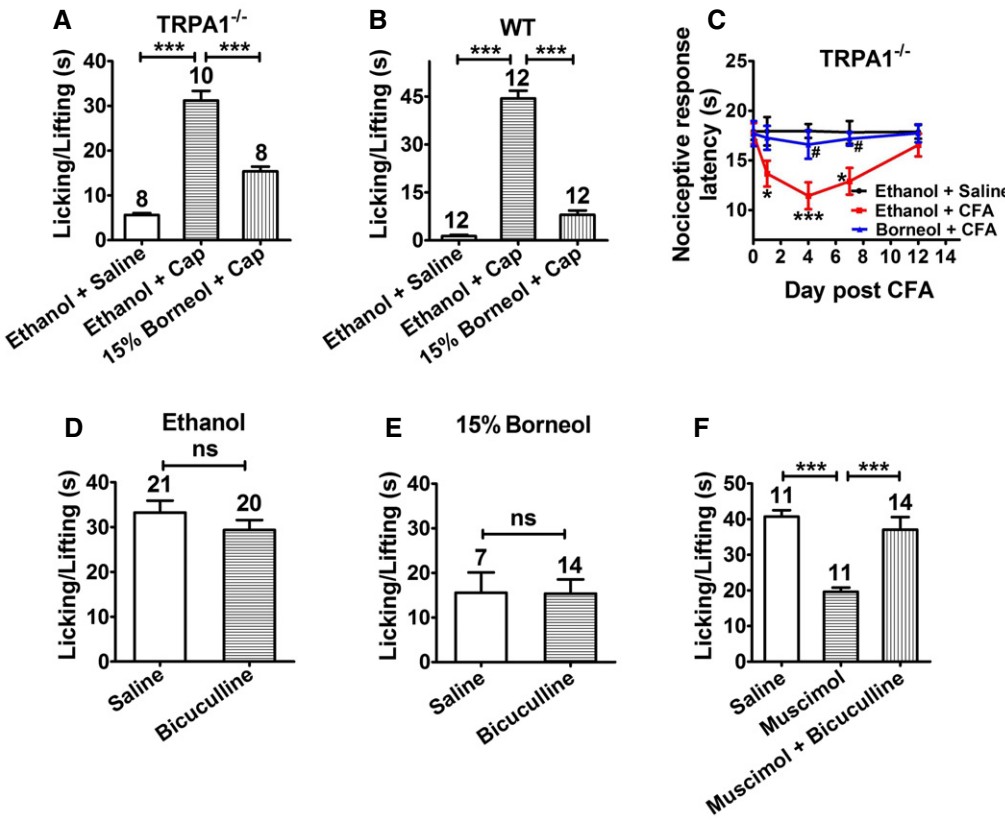

**Figure 3.  The analgesic effect of topical borneol in TRPA1$^{-/-}$ mice and wild-type (WT) mice with pharmacologically inhibited GABA$_A$.**

A, B    Quantification of the borneol effect on capsaicin (Cap)-induced nociceptive responses in TRPA1$^{-/-}$ mice (A) and WT mice (B). The experimental procedure was the same as that in Fig 2A.

C    Quantification of the borneol effect on CFA-induced thermal hyperalgesia in TRPA1$^{-/-}$ mice. The experimental procedure and statistical analysis were the same as that in Fig 2E. $n$ = 8 mice for each group.

D, E    Quantification of the effect of intrathecal injection of GABA$_A$ antagonist bicuculline on borneol-induced analgesia in the capsaicin model. After intrathecal injection of control saline or bicuculline in WT mice, ethanol (D) or 15% borneol (E) was applied to a hindpaw for a total of three times. After 10 min, 100 μM Cap was injected into the paw, and paw licking and lifting time was measured. Saline was used as a control to bicuculline, and ethanol was used as a control to borneol.

F    A positive control experiment of the intrathecal injection of bicuculline in WT mice. After intrathecal injection of control saline or of the GABA$_A$ agonist muscimol with or without bicuculline, 100 μM Cap was injected into the paw. Paw licking and lifting time was measured within 5 min. Saline was used as a control to muscimol and bicuculline.

Data information: In (A, B, D–F), the number of mice is indicated on top of each bar. Statistical significance was evaluated using two-tailed $t$-test. *$P < 0.05$; ***$P < 0.001$; #$P < 0.05$; ns indicates $P > 0.05$; the exact $P$-values are indicated in Appendix Table S1. All the data are presented as the mean ± standard error of the mean (SEM).

and EV3A). AMTB, a TRPM8-selective antagonist, completely inhibited borneol-induced currents (Fig 4I). Similarly, borneol also elicited intracellular Ca$^{2+}$ increases (Fig EV3B and C) and AMTB-sensitive currents (Fig EV3D) in HEK 293 cells expressing mouse TRPM8. These results together indicate that TRPM8 is a molecular target of borneol and that it is more sensitive to borneol than other already known targets, including TRPA1, GABA$_A$, and TRPV3.

## TRPM8 is the major mediator of topical borneol-induced analgesia in mice

We next investigated the role of TRPM8 in topical borneol-induced analgesia. Genetic ablation of TRPM8 substantially diminished the analgesic effect of topical borneol. In the capsaicin model, borneol suppressed the pain behavior by only 22% in TRPM8$^{-/-}$ mice compared with 85% in WT mice (compare Figs 5A and 3B).

Borneol-induced analgesia was also substantially attenuated in both phases of the formalin-induced pain responses in TRPM8$^{-/-}$ mice (Fig 5B). Topical borneol decreased the pain behavior by only 29% in the first phase and 4% in the second phase in TRPM8$^{-/-}$ mice compared with 53 and 91%, respectively, in WT mice (Fig 5B). In the CFA model, the analgesic effect of topical borneol was virtually absent in TRPM8$^{-/-}$ mice (Fig 5C and D). In agreement with the TRPM8 knockout experiments, pharmacological inhibition of TRPM8 also greatly reduced the analgesic effect of topical borneol. Intraplantar injection of AMTB (0.1 μg/μl) 15 min before topical application of borneol or ethanol did not affect capsaicin- or formalin-induced pain responses in mice but significantly reversed borneol-induced analgesia (Fig 5E and F). These results indicate that TRPM8 is a major mediator of topical borneol-induced analgesia in mice, and also show that borneol has no effect on the ability of mice to respond to painful stimuli, suggesting a specific analgesic effect.

    

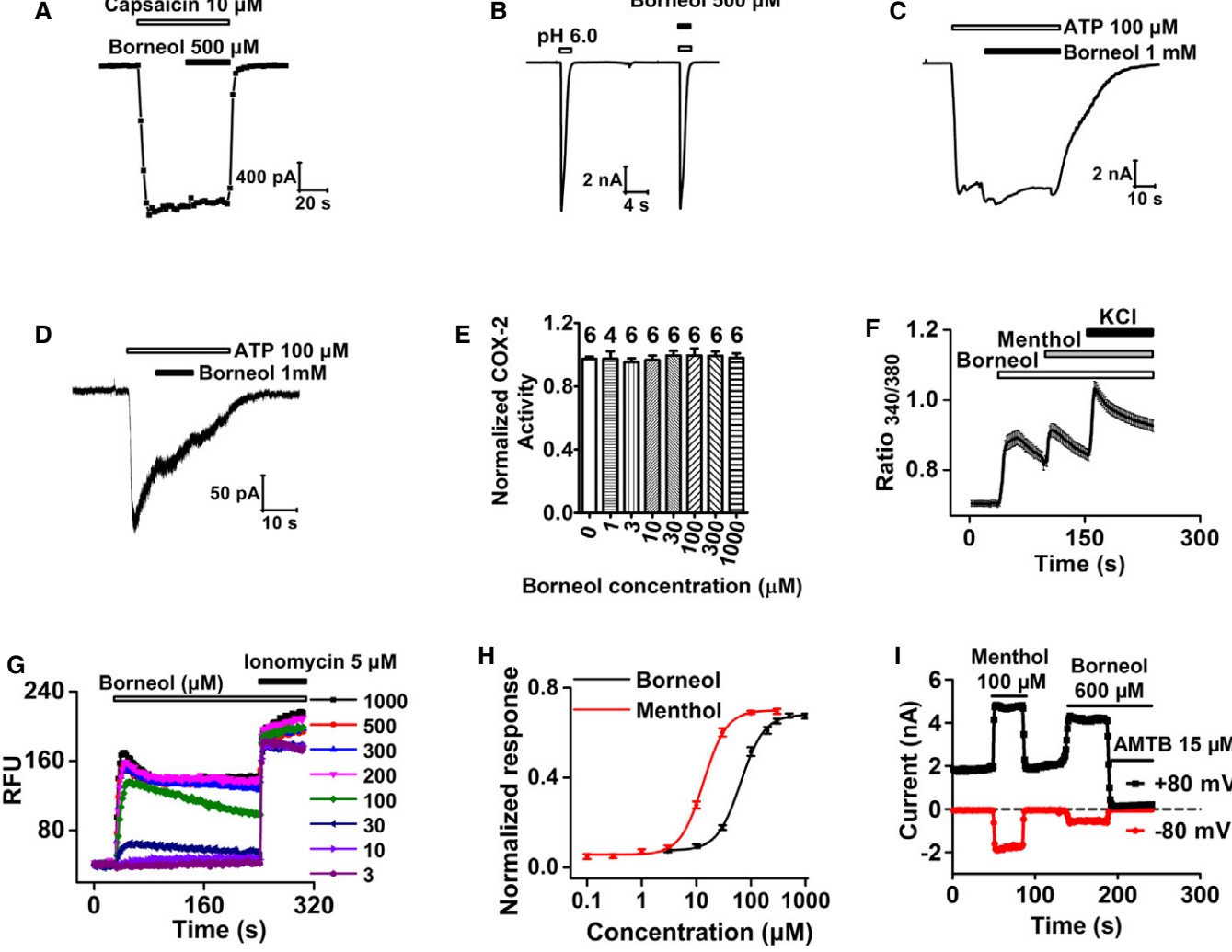

**Figure 4. The effects of borneol on common peripheral molecular targets in pain sensation.**

A–D  Representative whole-cell currents in HEK 293 cells expressing TRPV1 (A), ASIC3 (B), P2X2 (C), or P2X4 (D) in response to capsaicin (A), acidic pH (B), or ATP (C, D) in the absence of presence of borneol (*n* = 5 for each channel).

E  The effect of different concentrations of borneol on the enzymatic activity of human COX-2. The number of independent measurements is marked on top of each bar.

F  Averaged intracellular $Ca^{2+}$ increases in cultured mouse DRG neurons in response to consecutive applications of 200 μM borneol, 200 μM menthol, and 67 mM KCl. A total of 81 in 1,689 neurons from four mice were found to be borneol-sensitive and were included in the analysis.

G  Representative intracellular $Ca^{2+}$ signals in HEK 293 cells expressing human TRPM8 (hTRPM8) in response to different concentrations of borneol. After each application of borneol, $Ca^{2+}$ ionophore ionomycin was applied to calibrate $Ca^{2+}$ response. RFU: relative fluorescence unit.

H  Dose–response curves of the borneol- or menthol-induced increase in intracellular $Ca^{2+}$ in hTRPM8-expressing HEK 293 cells. Smooth curves are fit to the Hill equation with an $EC_{50}$ of 65 μM and a Hill coefficient of 2.0 for borneol (*n* = 15) and an $EC_{50}$ of 13 μM and a Hill coefficient of 2.0 for menthol (*n* = 6 at concentrations of 0.1, 0.3, and 1 μM; *n* = 9 at concentration of 3 μM; *n* = 13 at concentrations of 10, 30, 100, and 300 μM). The data were normalized to ionomycin-induced intracellular $Ca^{2+}$ increases.

I  Time course of menthol- and subsequently applied borneol-induced whole-cell currents in hTRPM8-expressing HEK 293 cells (*n* = 6).

Data information: All the data are presented as the mean ± standard error of the mean (SEM).

## Comparison of topical borneol and menthol in pain treatment

Menthol is a potent TRPM8 agonist and is well known for its analgesic effect, and a recent study showed that menthol-induced analgesia mainly occurred through TRPM8 (Liu *et al*, 2013). Thus, we compared the pharmacological effects of topical borneol and menthol in mice. Because borneol and menthol have very similar molecular weights

(154 vs. 156), the concentration of 15% was used for both agents in topical application. Borneol and menthol exhibited very similar analgesic effects on capsaicin- or formalin-induced pain behavior in mice (Fig 6A and B). However, genetic ablation of TRPM8 did not dampen the analgesic effect of topical menthol as effectively as that of borneol. In the capsaicin model, 15% menthol still caused 48% reduction in the pain behavior in TRPM8$^{-/-}$ mice, but 15% borneol caused only 22%

reduction in pain (Fig 6C). In the CFA model, 15% menthol-induced analgesia in TRPM8$^{-/-}$ mice was very similar to that in WT mice (compare Figs 6D with 2E), but 15% borneol-induced analgesia was virtually absent in TRPM8$^{-/-}$ mice (Fig 5D). These results suggest that borneol-induced analgesia mainly involves TRPM8, whereas menthol-induced analgesia involves not only TRPM8 but also other mechanisms. Previous studies reported that the opioid antagonist naloxone significantly blocked menthol- and menthol derivative-induced analgesia (Taniguchi *et al*, 1994; Galeotti *et al*, 2002; Liu *et al*, 2013), suggesting a downstream opioid mechanism. Indeed, in our study, intrathecal injection of naloxone significantly reversed topical menthol-induced analgesia in both WT and TRPM8$^{-/-}$ mice (Figs 6E and F, and EV4A). In contrast, intrathecal naloxone did not significantly affect topical borneol-induced analgesia (Fig 6G and H). In a parallel positive control experiment, naloxone effectively attenuated intrathecal morphine-induced analgesia in the capsaicin model (Fig EV4B). Interestingly, intrathecal injection of a selective group II metabotropic glutamate receptors (mGluRs) antagonist, LY341495, significantly diminished the analgesic effect of topical borneol (Fig 6I and J). This finding is consistent with a previous study showing that the analgesic effect of TRPM8 activation is mediated by group II/III mGluRs but not opioid receptors in the spinal cord (Proudfoot *et al*, 2006).

Numerous studies have reported that topical menthol causes cold allodynia and hyperalgesia in animals and humans (Hatem *et al*,

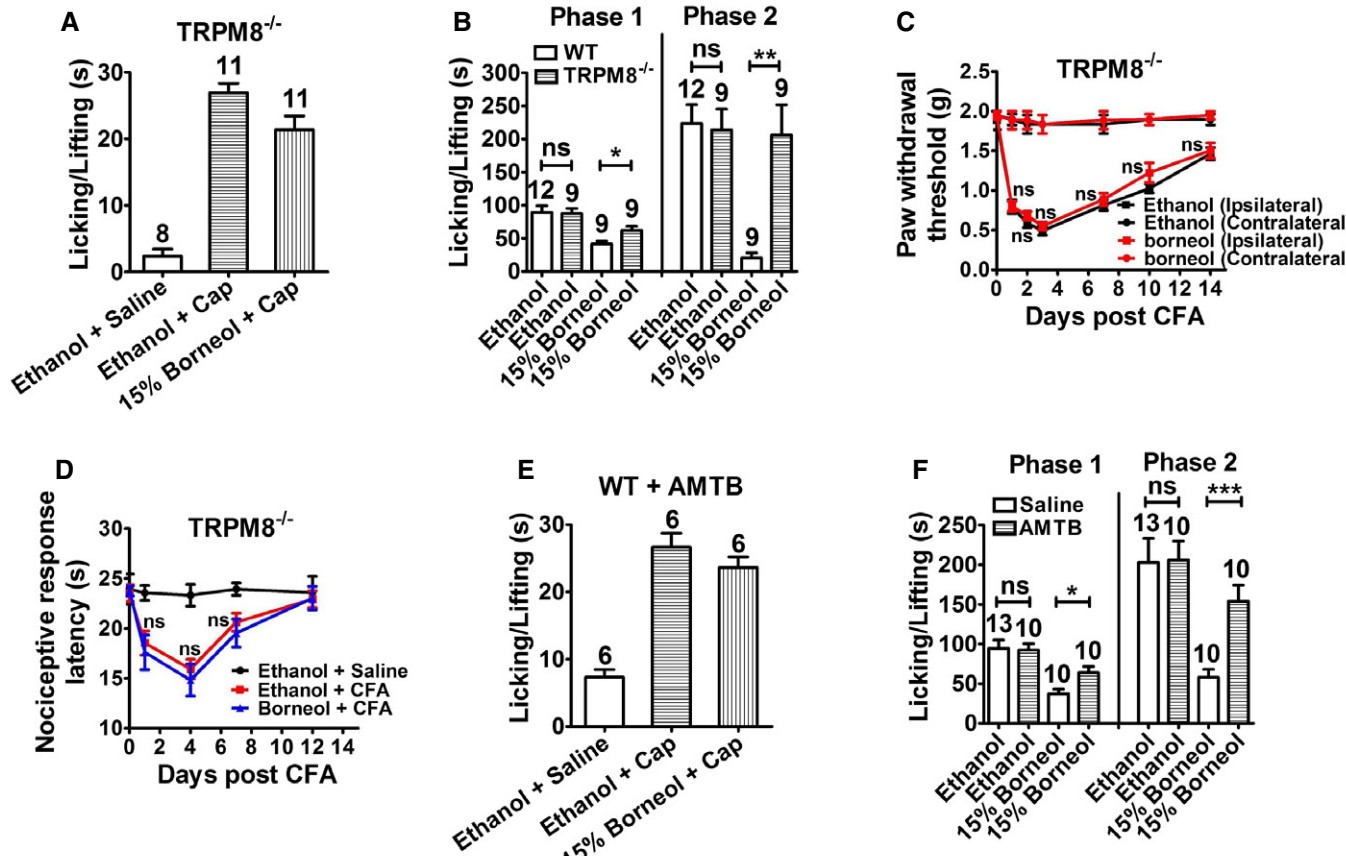

**Figure 5.  The effect of topical borneol in TRPM8$^{-/-}$ mice and WT mice with pharmacologically inhibited TRPM8.**

A   Quantification of the borneol effect on capsaicin (Cap)-induced nociceptive responses in TRPM8$^{-/-}$ mice. The experimental procedure was the same as that in Fig 2A.

B   Quantification of the borneol effect on formalin-induced biphasic pain responses in TRPM8$^{-/-}$ mice. The experimental procedure was the same as that in Fig 2C.

C   Quantification of the borneol effect on CFA-induced mechanical hyperalgesia in TRPM8$^{-/-}$ mice. The experimental procedure was the same as that in Fig 2D. The statistical analysis was performed between the groups of Borneol (Ipsilateral) and Ethanol (Ipsilateral). *n* = 8 mice for each group.

D   Quantification of the borneol effect on CFA-induced thermal hyperalgesia in TRPM8$^{-/-}$ mice. The experimental procedure was the same as that in Fig 2E. The statistical analysis was performed between the groups of Borneol + CFA (*n* = 12) and Ethanol + CFA (*n* = 10). *n* = 8 for the group of Ethanol + Saline.

E   Quantification of the borneol effect on Cap-induced nociceptive responses in WT mice with TRPM8 being inhibited pharmacologically. TRPM8 antagonist AMTB was injected into a hindpaw 15 min before borneol application. Subsequent experimental procedure was the same as that in Fig 2A.

F   Quantification of the borneol effect on formalin-induced biphasic pain responses in WT mice with or without TRPM8 being inhibited pharmacologically. Saline or AMTB was injected into a hindpaw 15 min before borneol application. 15% borneol was applied to the paw for a total of three times, and 1.5% formalin was injected into the paw after 10 min. Paw licking and lifting time in both phases was measured. Saline was used as a control to AMTB and ethanol was used as a control to borneol.

Data information: In (A, B, E, F), the number of mice is indicated on top of each bar. Statistical significance was evaluated using two-tailed *t*-test. *$P < 0.05$; **$P < 0.01$; ***$P < 0.001$; ns indicates $P > 0.05$; the exact *P*-values are indicated in Appendix Table S1. All the data are presented as the mean ± standard error of the mean (SEM).

    

2006; Green & Schoen, 2007; Altis *et al*, 2009; Klein *et al*, 2010). Thus, we examined whether borneol caused cold hypersensitivity in mice. The latency of the nociceptive response of mice was measured on a 0°C cold plate. Topical application of 15% menthol on hindpaw significantly decreased the latency of the nociceptive response to 0°C cold compared with ethanol treatment in naive mice (Fig 6K). TRPM8$^{-/-}$ mice showed attenuated nociception to 0°C cold (compare Fig EV5 and 6K), but 15% menthol still produced significant cold hypersensitivity in TRPM8$^{-/-}$ mice compared to ethanol (Fig EV5), suggesting that menthol-induced cold hypersensitivity has a TRPM8-independent mechanism. However, topical borneol did not significantly change the tolerance of mice to 0°C cold (Fig 6K). Similarly, in the CFA model, topical application of borneol on the mouse hindpaw did not generate significant nociceptive responses on a 10°C cold plate within a 200-s cutoff time (Fig 6L). In contrast, topical menthol-induced substantial nociceptive responses to 10°C cold in mice with a short latency (Fig 6L). These results indicate that topical borneol has a similar analgesic efficacy as menthol does but has a more restrictive effect and a safer profile.

## Discussion

Our randomized, double-blind, placebo-controlled clinical study demonstrated that a single application of 25% borneol for approximately 30–60 min produced a substantial reduction in postoperative pain in patients, validating the basis for the historical use of borneol as a topical analgesic in TCM. Topical borneol is significantly more effective than placebo and exhibits a clinical efficacy comparable to NSAID-containing topical analgesics. NSAIDs are the most studied topical analgesics and have a relative benefit of approximately 1.6–2.0 (Moore *et al*, 1998; Mason *et al*, 2004b,c); the relative benefit of borneol in our study was 1.89. The analgesic efficacy of borneol is also comparable or better than capsaicin-containing (relative benefit of approximately 1.4 or 1.5; Mason *et al*, 2004a) or menthol-containing topical analgesics (Higashi *et al*, 2010; Sundstrup *et al*, 2014). Notably, borneol was only applied for a short time in our study, although most topical analgesics are applied several times per day for several days to several weeks. It should be pointed out that our clinical study was a preliminary one and was limited to only a single dose and a particular pain condition. Future studies need to test different dosage forms, treatment protocols, and different pain conditions. It may also be advantageous to combine borneol with other agents targeting different mechanisms to achieve more complete analgesia.

The molecular and cellular mechanisms underlying borneol-induced analgesia have long stymied pharmacologists in TCM. In the past decade, GABA$_A$, TRPV3, and TRPA1 have been identified as the molecular targets of borneol (Granger *et al*, 2005; Vogt-Eisele *et al*, 2007; Takaishi *et al*, 2014; Sherkheli *et al*, 2015). However, TRPV3 activation contributes to pain hypersensitivity, and its antagonists have entered into clinical trials as analgesic agents (Nilius *et al*, 2014). Thus, it is unlikely that borneol-induced TRPV3 activation produces analgesia. One study reported that the GABA$_A$ receptor antagonist bicuculline abolished borneol-induced analgesia (Jiang *et al*, 2015), and another study hypothesized that TRPA1 channels accounted for borneol analgesia (Zhou *et al*, 2016). However, in both studies, borneol was intrathecally

administered at high concentrations. Therefore, whether those results explain the analgesic effect of topical borneol is questionable. Indeed, in our study, intrathecal bicuculline had no effect on topical borneol-induced analgesia, and genetic ablation of TRPA1 also showed that TRPA1 might only minimally contribute to the analgesic effect of topical borneol. We identified TRPM8 as the most sensitive molecular target of borneol discovered in mammals to date, with the EC$_{50}$ of 65 μM and a similar efficacy as menthol. Contradictorily, Vogt-Eisele *et al* (2007) showed that, even at 2 mM, borneol-induced response in TRPM8-expressing cells is < 10% of that induced by menthol, indicating that borneol has no significant effect on TRPM8. On the other hand, a recent paper also showed that borneol activated TRPM8 (Chen *et al*, 2016). However, the potency/efficacy of borneol activation of TRPM8 inferred from this study is significantly lower than that in our study. The reasons underlying the discord between these studies are unclear at present.

Genetic ablation or pharmacological inhibition of TRPM8 substantially diminished topical borneol-induced analgesia in mice, indicating that TRPM8 is the major mediator of borneol's analgesic effect. TRPM8 is a cold-activated channel and is extensively expressed by a subset of peripheral sensory neurons that diffusely innervate the skin, tongue, colon, and cornea (Dhaka *et al*, 2008; Parra *et al*, 2010; Harrington *et al*, 2011; Julius, 2013). A substantial fraction of these TRPM8-positive neurons do not express nociceptive markers, making them an anatomically and functionally unique subpopulation of primary afferent neurons (Dhaka *et al*, 2008; Julius, 2013). If excitation of peripheral nerve fibers causes analgesia, one possible explanation is that central mechanisms are involved. Glutamate is the main excitatory neurotransmitter in the synaptic transmission from primary afferents to spinal dorsal horn neurons. Its actions occur through ionotropic glutamate receptors and metabotropic glutamate receptors (mGluRs). The group II and III mGluRs are primarily localized on presynaptic terminals, and activation of these two mGluRs inhibits synaptic transmission and antagonizes pain (Gerber *et al*, 2000; Chiechio & Nicoletti, 2012). Our result that the group II mGluR antagonist LY341495 substantially reversed the analgesic effect of borneol suggests a central mechanism involving the mGluRs, which is consistent with the Proudfoot *et al* (2006) study that proposed a mode of TRPM8-mediated analgesia "in which Group II/III mGluRs would respond to glutamate released from TRPM8-containing afferents to exert an inhibitory gate control over nociceptive inputs".

It is worth noting that central opioid pathway did not contribute to TRPM8-mediated analgesia in both our and Proudfoot *et al*'s (2006) study, but our and previous studies showed that menthol- and a menthol derivative-induced analgesia is dependent on downstream opioid analgesic pathway (Taniguchi *et al*, 1994; Galeotti *et al*, 2002; Liu *et al*, 2013). These results suggest that at least a portion of the mechanisms underlying menthol-induced analgesia are TRPM8 independent. Menthol acts on many molecular targets (Swandulla *et al*, 1987; Haeseler *et al*, 2002; Karashima *et al*, 2007; Vogt-Eisele *et al*, 2007; Watt *et al*, 2008; Xiao *et al*, 2008; Zhang *et al*, 2008; Gaudioso *et al*, 2012; Hans *et al*, 2012; Pan *et al*, 2012; Ashoor *et al*, 2013), and some of these targets have been proposed as mediators of menthol-induced analgesia. Indeed, our results demonstrated that menthol still caused significant analgesia in TRPM8$^{-/-}$ mice and the opioid pathway was

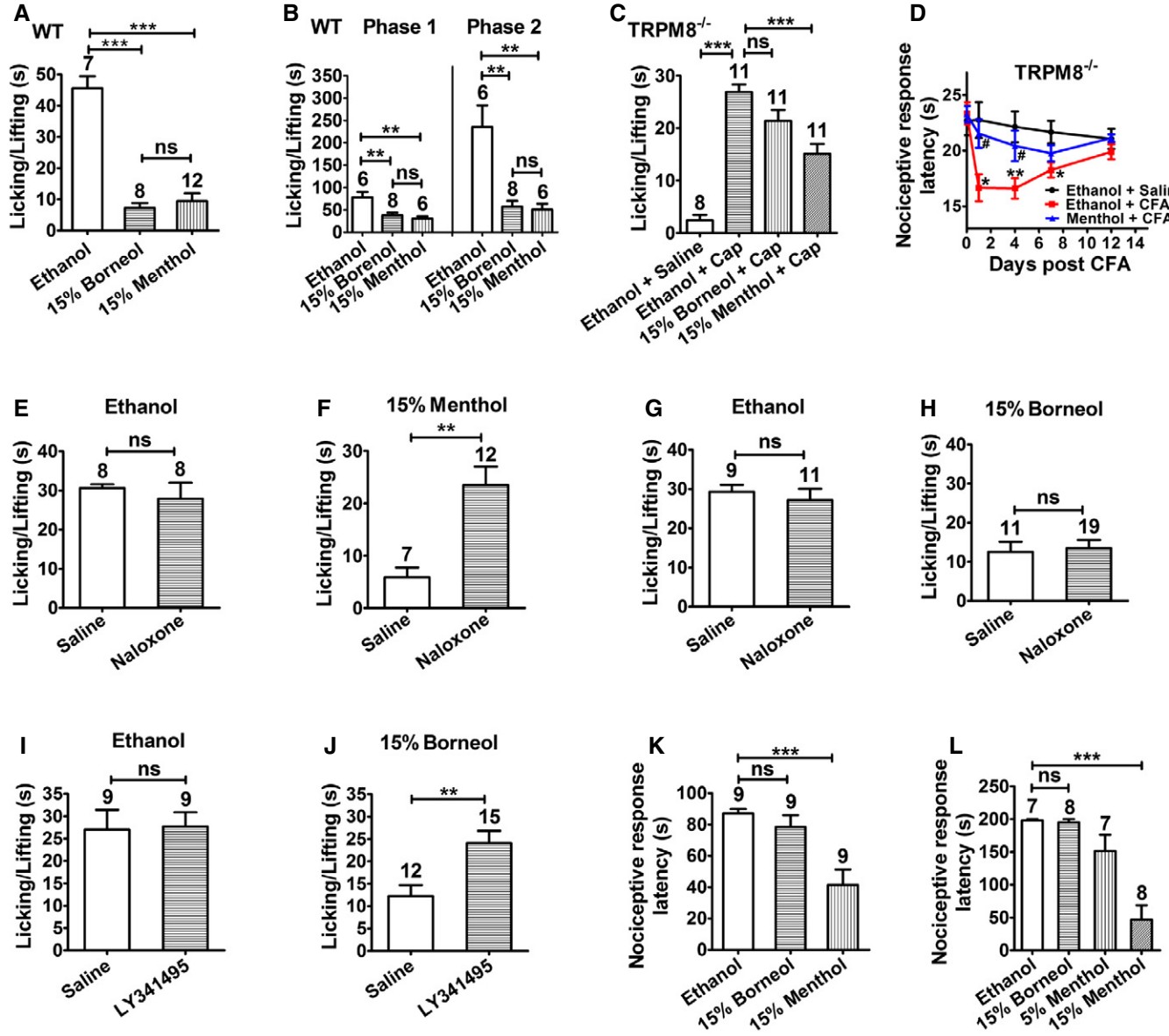

**Figure 6. Comparison of the effects of topical menthol and borneol in mice.**

A    Quantification of the effect of 15% borneol or 15% menthol on capsaicin-induced nociceptive responses in WT mice.

B    Quantification of the effect of 15% borneol or 15% menthol on formalin-induced biphasic pain responses in WT mice.

C    Quantification of the effect of 15% borneol or 15% menthol on capsaicin-induced nociceptive responses in TRPM8$^{-/-}$ mice.

D    Quantification of the menthol effect on CFA-induced thermal hyperalgesia in TRPM8$^{-/-}$ mice. The experimental procedure was the same as that in Fig 2E. The statistical analysis was performed between the groups of Ethanol + CFA ($n = 7$) and Ethanol + Saline ($n = 5$) with the statistical significance indicated by asterisks, or of Menthol + CFA ($n = 7$) and Ethanol + CFA ($n = 7$) with the statistical significance indicated by pound signs.

E–H  Quantification of the effect of intrathecal injection of opioid antagonist naloxone on borneol- or menthol-induced analgesia in the capsaicin model. After control saline or naloxone was intrathecally injected in WT mice, 15% menthol (F) or 15% borneol (H) was applied to a hindpaw for a total of three times. After 10 min, 100 µM Cap was injected into the paw, and paw licking and lifting time was measured within 5 min. Saline was used as a control to naloxone, and ethanol was used as a control to menthol (E) or borneol (G).

I, J  Quantification of the effect of intrathecal injection of group II mGluR antagonist LY341495 on borneol-induced analgesia in the capsaicin model. The experimental procedure was similar to that in naloxone experiments in (G and H).

K    Ethanol, borneol, or menthol was applied to both hindpaws of WT mice, and the nociceptive response latencies were measured in a cold plate test (0°C).

L    At 24 h after CFA injection, ethanol, borneol, or menthol was applied to the hindpaws, and the nociceptive response latencies were measured in a cold plate test (10°C).

Data information: In (A–C, E–L), the number of mice is indicated on top of each bar. Statistical significance was evaluated using two-tailed $t$-test (for two-group comparisons) or one-way analysis of variance (ANOVA) followed by Tukey's test (for multi-group comparisons). *$P < 0.05$; **$P < 0.01$; ***$P < 0.001$; #$P < 0.05$; ns indicates $P > 0.05$; the exact $P$-values are indicated in Appendix Table S1. All the data are presented as the mean ± standard error of the mean (SEM).

involved. However, in the Liu *et al* (2013) paper, menthol-induced analgesia was completely abolished in TRPM8$^{-/-}$ mice. This discord may be due to the differences in using menthol stereoisomers or the differences in administration routes of menthol or borneol between our and the Liu *et al* study. Our study also showed that although borneol and menthol exhibited very similar analgesic effects in mice, borneol had a more restrictive effect and a safer profile. Topical borneol did not cause significant noxious effects in both humans and mice in our study. In contrast, topical menthol applied to mouse hindpaw significantly promoted cold nociception, which is consistent with previous clinical studies in humans (Hatem *et al*, 2006; Green & Schoen, 2007; Altis *et al*, 2009). Moreover, our study showed that menthol can elicit cold hypersensitivity in a TRPM8-independent manner. The adverse effect of menthol may come from its multi-target effects, such as the activation of irritant channel TRPA1 (EC$_{50}$ = 28.4 ± 3.6 μM; Kwan *et al*, 2006; Xiao *et al*, 2008).

In summary, we provide evidence for the analgesic efficacy of topical borneol in humans through a rigorous clinical study and identify TRPM8 as the main molecular target mediating topical borneol-induced analgesia in mice. Our work suggests that borneol, alone or as an ingredient in combinatorial formulas, holds promise for future development of new topical analgesics.

# Materials and Methods

### Chemicals

Medical borneol was obtained from Shanghai Hongqiao Traditional Chinese Medicine Co., Ltd. (+)-Borneol (simply called borneol in this study), (±)-menthol (simply called menthol in this study), complete Freund's adjuvant, and capsaicin were purchased from Sigma-Aldrich. Formalin was purchased from Xilong Chemical Co. AMTB, naloxone, and LY341495 were obtained from TOCRIS. Bicuculline was obtained from Tokyo Chemical Industry. Muscimol was obtained from Enzo Life Sciences. ATP disodium salt was obtained from Sangon Biotech. Ionomycin was purchased from Cayman Chemical. Morphine hydrochloride was obtained from Northeast Pharm.

### Clinical study

This clinical study was performed as a randomized, double-blind, placebo-controlled study in adult female and male patients at the Shanghai Changzheng Hospital, which is affiliated with the Second Military Medical University in Shanghai, China. The study was approved by the Research Ethics Committee of the Shanghai Changzheng Hospital and conducted in accordance with the ethical principles of the Declaration of Helsinki. This study was registered in Chinese Clinical Trial Registry (ChiCTR) with registration number of ChiCTR-IOR-16009714. The patients provided written informed consent after verbal and written explanation of the study protocol.

In total, 126 patients, aged 18–85 years, were enrolled and allocated either to the control group or borneol group by computerized randomization. All of these patients were at the 2$^{nd}$ or 3$^{rd}$ day after spinal surgery and had a postoperative pain with a score of at least 30 mm on a 100-mm visual analog scale (VAS). Two patients were

excluded due to wound infection and a lack of clear consciousness, respectively. Two patients quit the study without providing a reason. In total, 122 patients completed the study and were included in the analysis. A hospital staff member prepared the 25% borneol solution and the placebo solution (ethanol), and labeled the solutions in advance. Hospital staff members who had no information regarding the clinical study and the solutions performed the treatment and recorded the pain scores in patients. Borneol solution or placebo was locally applied once to the skin around the sutured wound for approximately 30–60 min. The pain intensity was assessed by asking the patients to indicate the intensity of their current pain on a 100-mm VAS between 0 (no pain) and 100 (worst possible pain) before and after the treatment. An independent analyst analyzed the data.

### Animal experiments

Mice used for this study were maintained in the animal services facility of Kunming Institute of Zoology. All animal experiments were approved by the Animal Care and Use Committee at Kunming Institute of Zoology of Chinese Academy of Sciences, and the principles of laboratory animal care (NIH publication No. 86-23, revised 1985) were followed.

TRPA1$^{+/-}$ and TRPM8$^{+/-}$ mice were originally obtained from David Julius' lab and the Jackson Laboratory, respectively. TRPA1$^{-/-}$ and TRPM8$^{-/-}$ and littermate WT mice were generated from TRPA1$^{+/-}$ or TRPM8$^{+/-}$ mice, and were identified by genotyping PCR. 7- to 9-week-old mice were used (half male and half female). Animals were randomly selected and allocated to experimental groups, but no blinding was done. All experiments were performed at room temperature (approximately 23°C).

Before the measurement of capsaicin- or formalin-induced pain, mice were acclimated to a Plexiglas chamber for at least 30 min. Borneol, menthol, or ethanol was topically applied to mouse hindpaws by immersion of the paw in the solution for 1–2 s with a 10-min interval for a total of three times. At 10 min after the third application, freshly prepared 100 μM capsaicin solution (20-μl volume), 1.5% formalin solution (15-μl volume), or control saline was injected into plantar surface of the hindpaws to induce pain behavior, and the mice were immediately returned to the Plexiglas chamber and recorded using a digital video camera. The time mice spent licking or lifting the injected paw was counted within 5 min after capsaicin injection or within 5 min and 15–60 min after formalin injection. In the CFA model, 20 μl of CFA (1:1 emulsion of saline) was injected into mouse plantar surface to induce chronic inflammation in a single or both hindpaws. The mechanical and thermal hyperalgesia were studied within 2 weeks after the CFA injection. For measurement of the mechanical hyperalgesia, the mice were placed in a plastic cage with a wire mesh bottom and were allowed to acclimate until cage exploration and major grooming activities ceased. 10 min after application of borneol or ethanol, 50% paw withdrawal thresholds were assessed with von Frey filaments (North Coast Medical Inc.) using the up-down method (Chaplan *et al*, 1994). Thermal hyperalgesia in mice was evaluated using a hot/cold plate (Bioseb, Cold Hot Plate). After application of borneol, menthol, or ethanol to mouse hindpaws, mice were placed on a stainless steel plate with the setting temperatures of 52°C (30-s cutoff time), 0°C (90-s cutoff time), or 10°C (200-s cutoff time), and

the latency time for mice to exhibit nociceptive responses (licking hindpaw or jumping from the plate) was recorded.

Intrathecal injections were made at the L5–L6 intervertebral space in unanaesthetized mice with a Hamilton syringe. The proper insertion of the needle was verified by a quick flicking of the mouse's tail on entry of the needle. Bicuculline (15 ng), naloxone (30 ng), LY341495 (10 nM), or the respective control vehicle was intrathecally injected in a 5-μl volume immediately before topical application of borneol, menthol, or ethanol. Morphine (0.5 μg) with or without naloxone (30 ng) was intrathecally injected (5-μl volume) in the capsaicin model. Bicuculline (15 ng) was intrathecally injected (5-μl volume) 5 min before the intrathecal injection of muscimol (3 ng) in the capsaicin model.

### Primary DRG neuronal culture

Newborn WT or TRPM8$^{-/-}$ mice were euthanized following the guidelines of the Animal Care and Use Committee of Kunming Institute of Zoology, Chinese Academy of Sciences. Dorsal root ganglia were dissected, rinsed with Hank's buffer (Gibco), and digested in the same solution containing 1 mg/ml collagenase P (Roche Diagnostics) for 15–30 min at 37°C. Partially digested tissues were centrifuged at $200 \times g$ for 3 min, and the pellets were resuspended in 0.25% trypsin-EDTA (Gibco) and digested for an additional 5 min at 37°C. The digested ganglia were spun down, resuspended, and triturated with plastic pipette tips to release the neurons. The cells were filtered through a 70-μm cell strainer (Biologix), plated into 96-well plates, and cultured in DMEM/F-12 supplemented with GlutaMAX (Gibco), 10% serum (Gibco), and penicillin (100 U/ml)/streptomycin (0.1 mg/ml; Biological Industries). The calcium imaging experiment was performed after 48 h.

### HEK 293 cell culture and transfection

HEK 293 cells (American Type Culture Collection (ATCC), not tested for mycoplasma) or HEK 293 cells stably expressing TRPM8 channels were grown in DMEM (HyClone) plus 10% fetal bovine serum (Gibco) and penicillin (100 U/ml)/streptomycin (0.1 mg/ml; Biological Industries) with or without G418 sulfate (0.2 mg/ml, Gibco). HEK 293 cells were transiently transfected with plasmids using LipoD293 *In Vitro* DNA Transfection Reagent (SignaGen Laboratories) and used within 48 h.

### Intracellular Ca$^{2+}$ imaging

Intracellular calcium imaging of HEK 293 cells was performed using the FlexStation 3 microplate reader (Molecular Devices). Cells were plated in 96-well plates and loaded with the calcium-sensitive fluorescent dye Fluo-4 AM (10 μM) and the surfactant polyol Pluronic F-127 (0.02%; Molecular Probes) at 37°C for 1 h in a Ca$^{2+}$-free imaging solution. Subsequently, real-time fluorescence changes in cells upon the addition of a test compound were measured.

For intracellular calcium imaging of DRG neurons, the neurons were loaded with the fluorescent ion indicator Fura-2 AM (10 μM) and Pluronic F-127 (0.02%; Molecular Probes) at 37°C for 1 h in a Ca$^{2+}$-free imaging solution. The fluorescence ratios of F340/F380 were measured using a fluorescence microscopic system (Olympus Corporation, Sutter Instrument and Molecular Devices). The

standard imaging solution contained (in mM) 145 NaCl, 5 KCl, 1 MgCl$_2$, 2 CaCl$_2$, and 10 HEPES, pH 7.4.

### Electrophysiology

All experiments were performed at room temperature (approximately 23°C). For patch-clamp recordings, pipettes were fabricated and fire-polished to resistances of 2~3 MΩ for whole-cell recording. The whole-cell currents of TRPM8 or TRPV1 were elicited in HEK 293 cells by 500-ms voltage ramps from −100 to +100 mV at a frequency of 0.5 Hz with a holding potential of 0 mV. The whole-cell currents of ASIC3, P2X2, and P2X4 were recorded in gap-free mode with the holding potential of −60 mV. Currents were amplified using an Axopatch 200B patch-clamp amplifier and digitized with Digidata 1440A (Molecular Devices). Currents were low-pass filtered at 2 kHz and sampled at 10 kHz. pCLAMP software (Molecular Devices) was used for data acquisition and analysis. The extracellular solution contained (in mM) 150 NaCl, 1 MgCl$_2$, and 10 HEPES (pH 7.4). For TRP channel recording, the standard intracellular solution contained (in mM) 150 NaCl, 1 MgCl$_2$, 1 EGTA, and 10 HEPES (pH 7.4). For recording ASIC3, P2X2, and P2X4, the intracellular solution contained (in mM) 140 KCl, 5 EGTA, and 10 HEPES (pH 7.4). Extracellular solutions containing borneol, menthol, capsaicin, or ATP were prepared immediately before the experiments.

### Cyclooxygenase-2 functional assay

The effect of borneol on the activity of human COX-2 was determined using the COX-2 (human) Inhibitor Screening Assay Kit (Cayman Chemical) according to the manufacturer's instructions.

### Statistics

For the clinical study, pre-study calculations suggested that 49 patients per group were required to detect a 10-mm difference in the average VAS change between the borneol group and placebo group with an SD of 15 at the 0.05 significant level and 90% power (GPower, Universität Düsseldorf, Germany). In fact, 62 patients in the borneol group and 60 patients in the placebo group finished the assessment and were included in the statistical analysis in this study, and the observed SD was 15 and 17 in the two groups, respectively, with a 97% power to detect the difference in the average VAS change between the borneol and placebo groups. Data normality was assessed with the Shapiro–Wilk test, and the equality of variances for the two data groups was determined with the Levene's test. The chi-square test was used to compare the gender or the percentage of patients with a ≥ 50% reduction in pain score in the borneol and placebo groups. The Mann–Whitney $U$-test was conducted to determine significant differences between groups in age, VAS scores before treatment, VAS scores after treatment, the absolute change in VAS scores, and the percentage change in VAS scores. Relative benefit with 95% CI was calculated using the fixed-effects model (Morris & Gardner, 1988). The data are expressed as the mean (SD, 95% CI) unless otherwise specified. $P < 0.05$ was considered significant.

For the animal study, we used sample/animal sizes that were deemed suitable for statistics and were similar to other studies in the field. Statistical significance was evaluated using two-tailed $t$-test

**The paper explained**

**Problem**

(+)-Borneol has been used as a topical analgesic for millennia in traditional Chinese medicine, but its clinical efficacy lacks stringent evidence-based clinical studies and verifiable scientific mechanism.

**Results**

We demonstrate in a human clinical study that topical application of borneol significantly relieves pain. Using mouse models of pain, we identify the TRPM8 channel as a molecular target of borneol. Topical borneol-induced analgesia is nearly exclusively mediated by TRPM8 and involves a downstream glutamatergic mechanism in the spinal cord. We further show mechanistic differences between borneol- and menthol-induced analgesia and demonstrate that borneol exhibits advantages over menthol as a topical analgesic.

**Impact**

Our work provides rigorous clinical evidence for the analgesic efficacy of topical borneol in humans and elucidates its underlying mechanism in mouse models. This study may pave the way for future development of new borneol-based topical analgesics.

(for all two-group comparisons) or one-way analysis of variance (ANOVA) followed by Tukey's test (for multi-group comparisons). The data are presented as the mean ± standard error of the mean (SEM), and $P < 0.05$ was considered statistically significant.

**Expanded View** for this article is available online.

## Acknowledgements

We thank David Julius of the University of California San Francisco for providing the TRPA1$^{+/-}$ mice and Zhuan Zhou of Peking University for shipping the TRPA1$^{+/-}$ mice to us. We also thank members of the Ion Channel Research and Drug Development Center at Kunming Institute of Zoology for discussions. This work was supported by grants from the National Natural Science Foundation of China (81302865), Yunnan Applied Basic Research Projects (2013FB074), Youth Innovation Promotion Association of the Chinese Academy of Sciences, West Light Foundation of the Chinese Academy of Sciences, and Key Research Program of the Chinese Academy of Sciences (KJZD-EW-L03) to Shu Wang; grants from the Top Talents Program of Yunnan Province (2011HA012), National Basic Research Program of China (2014CB910301), High-level Overseas Talents of Yunnan Province, Yunnan Major Science and Technology Project (2015ZJ002), and the National Natural Science Foundation of China (NSFC) (31370821) to Jian Yang; grants from the High-level Overseas Talents of Yunnan Province and the National Basic Research Program of China (2014CB910301) to Ming Zhou; and a grant from the High-level Overseas Talents of Yunnan Province to Jianmin Cui.

## Author contributions

SW and JY conceived the study and designed all of the biological experiments, with inputs and discussion from MZ and JC, DZ and JH performed the animal experiments with assistance from DS and ZX, DZ, QJ, and DS performed the electrophysiology and Ca$^{2+}$ imaging experiments in HKE 293 cells. DZ and JH performed cell culture and Ca$^{2+}$ imaging of DRG neurons. JH performed the functional assay of COX-2. DZ, QJ, JH, DS, and SW analyzed the data from biological experiments. JX, DZ, and WX designed and supervised the clinical study, with inputs and discussion from SW and JY, HS analyzed the clinical data. SW, JY, and MZ wrote the manuscript, with inputs from all authors.

## Conflict of interest

The authors declare that they have no conflict of interest.

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
