## [Review Process File · EMBO Molecular Medicine]

A clinical and mechanistic study of topical borneol-induced analgesia

Shu Wang, Dan Zhang, Jinsheng Hu, Qi Jia, Wei Xu, Deyuan Su, Hualing Song, Zhichun Xu, Jianmin Cui, Ming Zhou, Jian Yang and Jianru Xiao

*Corresponding authors: Shu Wang, Chinese Academy of Sciences
Jian Yang, Columbia University
Jianru Xiao, The Second Military Medical University*

Review timeline:

Submission date:	06 November 2016
Editorial Decision:	12 December 2016
Revision received:	16 February 2017
Editorial Decision:	28 February 2017
Revision received:	03 March 2017
Accepted:	08 March 2017

Transaction Report:

Editor: Céline Carret

1st Editorial Decision

12 December 2016

Thank you for the submission of your manuscript to EMBO Molecular Medicine. We have now heard back from the three referees whom we asked to evaluate your manuscript.

Although the referees find the study to be of interest, you will see from the comments pasted below that they also suggest a number of additions, including experimental to strengthen the data and provide more molecular insights.

We would welcome the submission of a revised version for further consideration and depending on the nature of the revisions, this may be sent back to the referees for another round of review.

Please note that it is EMBO Molecular Medicine policy to allow only a single round of revision and that, as acceptance or rejection of the manuscript will depend on another round of review, your responses should be as complete as possible.

I look forward to receiving your revised manuscript.

***** Reviewer's comments *****

Referee #1 (Remarks):

The authors tested the herb used in traditional Chinese medicine (TCM) Bingpian that is >96% (+)-borneol as a topical analgesic in double blind placebo controlled clinical study, for the effect of a single dose applied topically for 30-60 min to patients with postoperative pain. Having found beneficial effects in both clinical trial and study of mouse models, they test for its effect on various ion channels expressed in HEK 293 cells, to identify TRPM8 as the candidate target. They then showed that the analgesic effects are absent in TRPM8 knockout mice. Their comparison with the menthol actions indicate that TRPM8 channel activation by borneol likely involves central action of mGluRII on presynaptic terminals, but are not dependent on opioid receptor signaling.

These findings are interesting and well documented by their experiments. For the TRPM8 channel activation by borneol shown in Figure 4i, it will be important to show quantification of multiple recordings as well as the ability of TRPM8 channel antagonists to block the borneol effect.

Referee #2 (Comments on Novelty/Model System):

No issue

Referee #2 (Remarks):

In this manuscript, S. Wang et al. first examined the therapeutic effect of borneol, the main ingredient of traditional Chinese medicine (TCM), pingpian, in patients with postoperative pain, and then investigated its molecular mechanism in antinociception using mouse models. The clinical data clearly demonstrate the benefit of borneol in suppressing pain at a single dose topical application. The behavior studies with mice and single cell functional assays reveal that borneol exerts analgesic effect through activation of TRPM8, a cold sensitive channel expressed in a small fraction of primary sensory neurons. TRPM8 is best known for its sensitivity to cool temperature and menthol. However, the authors found borneol to be more specific than menthol in terms of its nearly pure TRPM8-dependent mechanism of analgesic action and the lack of detectable cold nociception. Overall, the study is very interesting. Given that pain management represents a major challenge of modern medicine and growing awareness of the benefits of TCM herbs, this study is of significant values both at the clinical side and in basic science with mechanistic insights. The work was done very well and the paper is clearly written.

Major point:

While the authors demonstrate clearly that menthol may also exerts analgesic effect through activation of opioid receptors, the differential mechanism by which only menthol, but not borneol, caused cold allodynia and hyperalgesia in the CFA model was not explored. The authors suspected that these might be caused by the non-specific action of menthol on other targets, for example TRPA1. However, there is no data suggesting that these added effects of menthol were independent of TRPM8, as all experiments were done using wild type mice. Can naloxone prevent menthol from reducing capsaicin-induced pain also in TRPM8^{-/-} mice? Will menthol cause cold allodynia and hyperalgesia in TRPM8^{-/-} mice? A clear answer to this question will support the argument that the added effects of menthol, as compared to borneol, resulted from TRPM8-independent actions. This is important because previously studies have suggested the involvement TRPM8 in cold allodynia and both nociceptive and non-nociceptive DRG neurons may express TRPM8. As this point, whether these actions were due to TRPA1 or something else may not be that important.

Minor points:

- 1) Salicylates are often also considered as NSAIDs. Having them separately listed with NSAIDs (Page 4) is a little odd. In the same sentence, would "anesthetics" refer just "local anesthetics" here?
- 2) "greater improvements in pain" (Page 8) could mean either more pain or less pain. Do you mean pain suppression, pain control, or pain management?
- 3) How was the analgesic efficacy of 1.89 determined for borneol? What is the analgesic efficacy of menthol-containing topical analgesics (Page 15)?
- 4) In Fig. 4i, there is a blip in the middle of borneol treatment. Is this a nonspecific effect?
- 5) Fig. 4 legend, line 5, "is marked on top OF each bar".
- 6) Fig. 6g top label, "Ethonal" should be "Ethanol".

Referee #3 (Remarks):

This manuscript by Wang et. al aims to investigate the clinical analgesic effects and the underlying mechanisms of Borneol, a herbal compound that has been used in clinical applications for more than 2,000 years! Overall, this is an interesting story with some nicely-presented high-quality results ranging from well-controlled clinical studies, knockout mice, pain behavioral models, and imaging- or electrode-based physiological assays. Although identification of TRPM8 was recently reported to be activated by Borneol (Chen et al 2016), the significance of the current study was only slightly compromised, due to the clean results that the authors have obtained from the behavioral tests using TRPM8 knockout mice as negative controls. The manuscript could be improved if the authors could address following comments in the revision:

1. It would be nice to show the AMTB controls in Fig. 3a + Fig 5a. The prediction was that AMTB would completely abolish the capsaicin-induced licking in WT, but not TRPM8 KO mice.
2. The authors used mouse pain models to study the analgesia mechanisms of Borneol, and the channel assays were conducted on human TRPM8. Are there any species differences?
3. The authors used two different concentration units of drugs: % in the behavior tests and molar concentration for in vitro assays. It would be helpful if conversions could be somehow indicated.
4. In page12, it was stated that none of the neurons responded to Borneol, confirming that TRPM8 was the mediator. Likewise, it was stated in page 13 that Borneol had no effect on the locomotion of the mice. I was not able to find these results.
5. There were too many traces in Fig. 4f. AMTB could be a good control here.

1st Revision - authors' response

16 February 2017

We thank the reviewers for their insightful critiques. We address their concerns and questions in detail below. In revising the manuscript, we have performed new experiments suggested by the reviewers – the results of which support the conclusions. We also have rewritten some parts of the text and redrawn figures accordingly. We hope we have satisfactorily addressed the reviewers' concerns and questions.

Referee #1 (Remarks):

The authors tested the herb used in traditional Chinese medicine (TCM) Bingpian that is >96% (+)-borneol as a topical analgesic in double blind placebo controlled clinical study, for the effect of a single dose applied topically for 30-60 min to patients with postoperative pain. Having found beneficial effects in both clinical trial and study of mouse models, they test for its effect on various ion channels expressed in HEK 293 cells, to identify TRPM8 as the candidate target. They then showed that the analgesic effects are absent in TRPM8 knockout mice. Their

comparison with the menthol actions indicate that TRPM8 channel activation by borneol likely involves central action of mGluRII on presynaptic terminals, but are not dependent on opioid receptor signaling.

These findings are interesting and well documented by their experiments. For the TRPM8 channel activation by borneol shown in Figure 4I, it will be important to show quantification of multiple recordings as well as the ability of TRPM8 channel antagonists to block the borneol effect.

We quantified borneol-induced currents in hTRPM8-expressing cells (Fig. 4I and Fig. EV3A in the revised manuscript) and showed that AMTB, a TRPM8-selective antagonist, completely blocked the borneol effect (Fig. 4I).

Referee #2 (Remarks):

In this manuscript, S. Wang et al. first examined the therapeutic effect of borneol, the main ingredient of traditional Chinese medicine (TCM), pingpian, in patients with postoperative pain, and then investigated its molecular mechanism in antinociception using mouse models. The clinical data clearly demonstrate the benefit of borneol in suppressing pain at a single dose topical application. The behavior studies with mice and single cell functional assays reveal that borneol exerts analgesic effect through activation of TRPM8, a cold sensitive channel expressed in a small fraction of primary sensory neurons. TRPM8 is best known for its sensitivity to cool temperature and menthol. However, the authors found borneol to be more specific than menthol in terms of its nearly pure TRPM8-dependent mechanism of analgesic action and the lack of detectable cold nociception. Overall, the study is very interesting. Given that pain management represents a major challenge of modern medicine and growing awareness of the benefits of TCM herbs, this study is of significant values both at the clinical side and in basic science with mechanistic insights. The work was done very well and the paper is clearly written.

Major point:

While the authors demonstrate clearly that menthol may also exerts analgesic effect through activation of opioid receptors, the differential mechanism by which only menthol, but not borneol, caused cold allodynia and hyperalgesia in the CFA model was not explored. The authors suspected that these might be caused by the non-specific action of menthol on other targets, for example TRPA1. However, there is no data suggesting that these added effects of menthol were independent of TRPM8, as all experiments were done using wild type mice. Can naloxone prevent menthol from reducing capsaicin-induced pain also in TRPM8^{-/-} mice? Will menthol cause cold allodynia and hyperalgesia in TRPM8^{-/-} mice? A clear answer to this question will support the argument that the added effects of menthol, as compared to borneol, resulted from TRPM8-independent actions. This is important because previously studies have suggested the involvement TRPM8 in cold allodynia and both nociceptive and non-nociceptive DRG neurons may express TRPM8. As this point, whether these actions were due to TRPA1 or something else may not be that important.

We performed new experiments and the results showed that intrathecal naloxone significantly reversed the analgesic effect of topical menthol in TRPM8^{-/-} mice (Fig. EV4A in the revised manuscript), which suggests that the central opioid pathway contributes to TRPM8-independent analgesia caused by menthol. Genetic ablation of TRPM8 does attenuate the cold nociception in mice (compare Fig. 6K and Fig. EV5 in the revised manuscript), but menthol still caused significant cold hypersensitivity compared with ethanol treatment in TRPM8^{-/-} mice (Fig. EV5). These results indicate that menthol can cause pharmacological and pathological responses in a TRPM8-independent manner.

Minor points:

1) Salicylates are often also considered as NSAIDs. Having them separately listed with NSAIDs (Page 4) is a little odd. In the same sentence, would "anesthetics" refer just "local anesthetics" here?

We deleted salicylates and changed “anesthetics” to “local anesthetics” in the revised manuscript.

2) "greater improvements in pain" (Page 8) could mean either more pain or less pain. Do you mean pain suppression, pain control, or pain management?

We changed “greater improvements in pain” to “greater pain relief” in the revised manuscript.

3) How was the analgesic efficacy of 1.89 determined for borneol? What is the analgesic efficacy of menthol-containing topical analgesics (Page 15)?

Relative benefit estimate with 95% CIs was calculated using the fixed effects model. We indicated this and cited relevant paper in the Statistics section. To our knowledge, the relative benefit values of menthol-containing topical analgesics have never been reported, so we do not know.

4) In Fig. 4i, there is a blip in the middle of borneol treatment. Is this a nonspecific effect?

It is a nonspecific effect. We performed new recordings and replaced Fig. 4i with new data.

5) Fig. 4 legend, line 5, "is marked on top OF each bar".

We corrected the mistake.

6) Fig. 6g top label, "Ethonal" should be "Ethanol".

We corrected the mistake.

Referee #3 (Remarks):

This manuscript by Wang et. al aims to investigate the clinical analgesic effects and the underlying mechanisms of Borneol, a herbal compound that has been used in clinical applications for more than 2,000 years! Overall, this is an interesting story with some nicely-presented high-quality results ranging from well-controlled clinical studies, knockout mice, pain behavioral models, and imaging- or electrode-based physiological assays. Although identification of TRPM8 was recently reported to be activated by Borneol (Chen et al 2016), the significance of the current study was only slightly compromised, due to the clean results that the authors have obtained from the behavioral tests using TRPM8 knockout mice as negative controls. The manuscript could be improved if the authors could address following comments in the revision:

1. It would be nice to show the AMTB controls in Fig. 3a + Fig 5a. The prediction was that AMTB would completely abolish the capsaicin-induced licking in WT, but not TRPM8 KO mice.

AMTB is a TRPM8-selective antagonist. Thus, the prediction is that AMTB would have no effect on capsaicin-induced pain in WT mice, but antagonize the analgesic effect of borneol. We performed new experiments in the capsaicin model. AMTB had no significant effect on capsaicin-induced pain in WT mice (Fig. 5E in the revised manuscript), but significantly inhibited the analgesic effect of borneol (Fig. 5E), which is consistent with the results obtained from the TRPM8 KO mice (Fig. 5A).

2. The authors used mouse pain models to study the analgesia mechanisms of Borneol, and the channel assays were conducted on human TRPM8. Are there any species differences?

We performed Ca²⁺ imaging and patch-clamp recording on mouse TRPM8-expressing HEK 293 cells (Fig. EV3B, C, D), and borneol showed similar effects on mouse TRPA1.

3. The authors used two different concentration units of drugs: % in the behavior tests and molar concentration for in vitro assays. It would be helpful if conversions could be somehow indicated.

The reason for using % in the behavior tests is that % is mostly used for topical analgesics due to the high concentration of active ingredients. The conversion was shown when molar concentration of borneol was first used in the Results section in revised manuscript (page 12, line 5).

4. In page 12, it was stated that none of the neurons responded to Borneol, confirming that TRPM8 was the mediator. Likewise, it was stated in page 13 that Borneol had no effect on the locomotion of the mice. I was not able to find these results.

In the revised manuscript, Fig. EV1 demonstrates that none of the neurons from TRPM8 KO mice responded to borneol.

Indeed, the statement that “borneol has no effect on the locomotion of the mice” is inappropriate. What we really wanted to say is that topical borneol does not affect the ability of mice to respond to noxious stimuli, because TRPM8 KO mice showed normal nociception even after topical application of borneol (Fig. 5). We changed “borneol has no effect on the locomotion of the mice” to “borneol has no effect on the ability of mice to respond to painful stimuli” in the revised manuscript.

5. There were too many traces in Fig. 4f. AMTB could be a good control here.

The traces in Fig. 4F are averaged in the revised manuscript. AMTB is a TRPM8-selective antagonist and completely inhibited borneol-induced TRPM8 activation (Fig. 4I in the revised manuscript). A control experiment was done using DRG neurons from TRPM8 KO mice (Fig. EV1).

2nd Editorial Decision

28 February 2017

Thank you for the submission of your revised manuscript to EMBO Molecular Medicine. We have now received the enclosed reports from the referees that were asked to re-assess it. As you will see the reviewers are now supportive and I am pleased to inform you that we will be able to accept your manuscript pending the following final editorial amendments:

1) p-values: please indicate in legends exact n= and exact p= values, not a range. Some people found that to keep the figures clear, providing a supplemental table with all exact p-values was preferable. You are welcome to do this if you want to but as you do not have any Appendix file, it might be easier to only add the exact p-values within the legends.

2) please provide a clinical trial accession number within the manuscript and in the Authors checklist.

Please submit your revised manuscript within two weeks. I look forward to seeing a revised form of your manuscript as soon as possible.

***** Reviewer's comments *****

Referee #2 (Remarks):

The authors have addressed my concerns with new experiments confirming that menthol elicits TRPM8-independent behavioral responses using TRPM8 KO mice. The results look good and are consistent with the predictions based on the data included in the previous version and the authors' interpretation of these data.

Referee #3 (Remarks):

The authors have satisfactorily addressed all my concerns.

1) p-values: please indicate in legends exact n= and exact p= values, not a range. Some people found that to keep the figures clear, providing a supplemental table with all exact p-values was preferable. You are welcome to do this if you want to but as you do not have any Appendix file, it might be easier to only add the exact p-values within the legends.

We put all p-values in Appendix Table S1, and the n-values have been indicated in the figures or figure legends.

2) please provide a clinical trial accession number within the manuscript and in the Authors checklist.

This study was registered in Chinese Clinical Trial Registry (ChiCTR) with registration number of ChiCTR-IOR-16009714 which has been indicated in the manuscript and checklist.

Corresponding Author Name: Jian Yang

Journal Submitted to: EMBO molecular medicine

Manuscript Number: EMM-2016-07300